

**Deciphering Isoprene Variability Across Dozen of Chinese and Overseas Cities Using**
**Deep Transfer Learning**
Song Liu[1], Xiaopu Lyu[2]*, Fumo Yang[1], Zongbo Shi[3], Xin Huang[4], Tengyu Liu[4], Hongli Wang[5],
Mei Li[6], Jian Gao[7], Nan Chen[8], Guoliang Shi[9], Yu Zou[10], Chenglei Pei[11], Chengxu Tong[3], Xinyi
Liu[1], Li Zhou[1], Alex B. Guenther[12], and Nan Wang[1]*
[1]College of carbon Neutrality Future Technology, Sichuan University, Chengdu 610065, China.
[2]Department of Geography, Hong Kong Baptist University, Hong Kong 000000, China.
[3]School of Geography, Earth and Environmental Sciences, University of Birmingham,
Birmingham B15 2TT, UK.
[4]School of Atmospheric Sciences, Nanjing University, Nanjing 210023, China.
[5]State Environmental Protection Key Laboratory of Formation and Prevention of Urban Air
Pollution Complex, Shanghai Academy of Environmental Sciences, Shanghai, 200233, China.
[6]College of Environment and Climate, Institute of Mass Spectrometry and Atmospheric
Environment, Guangdong Provincial Engineering Research Center for On-line Source
Apportionment System of Air Pollution, Jinan University.
[7]Chinese Research Academy of Environmental Sciences, Beijing 100012, China.
[8]Research Centre for Complex Air Pollution of Hubei Province, Wuhan 430078, China.
[9]State Environmental Protection Key Laboratory of Urban Ambient Air Particulate Matter
Pollution Prevention and Control, Tianjin Key Laboratory of Urban Transport Emission
Research, College of Environmental Science and Engineering, Nankai University, Tianjin
300350, P. R. China.
[10]Institute of Tropical and Marine Meteorology, China Meteorological Administration,
Guangzhou, China.
[11]Guangzhou Sub-branch of Guangdong Ecological and Environmental Monitoring Center,
Guangzhou 510006, China.
[12]Department of Earth System Science, University of California, Irvine, California, USA.
Corresponding author: Xiaopu Lyu (xiaopu_lyu@hkbu.edu.hk); Nan Wang
(nan.wang@scu.edu.cn)





**Key Points:**
• An explainable deep transfer learning framework was developed to predict
isoprene concentrations and their variations.
• Different drivers accounted for historical trends of isoprene concentrations in
Hong Kong and London from 1990 to 2023.
• Reducing nitrogen oxides would alleviate ozone pollution driven by rising
temperatures and isoprene levels in the warming climate.





**Abstract**

Isoprene, the globally most abundant volatile organic compound, significantly impacts air quality. Determining isoprene concentration variations and their drivers is a persistent challenge. Here, we developed a robust machine learning framework to simulate isoprene concentrations, without requiring localized emission inventories and explicit chemistry. Temperature, radiation, and surface pressure were the primary drivers of short-term isoprene variations across Chinese cities. On climatic timescales, urban greenspace expansion and climate warming drove isoprene increases by 341 pptv in Hong Kong during 1990–2023, but traffic emission reductions in London counteracted the isoprene rise that climate warming would have otherwise caused (-755 pptv vs. +31 pptv). Driven by rising temperatures and isoprene levels, ozone would increase by up to 1.7-fold by 2100 under the high-emission scenario. However, ambitious reduction in nitrogen oxides would alleviate this growth to 1.2-fold. The study has the potential to inform air quality management in a warming climate.



## 1 Introduction

Isoprene is the most abundant non-methane volatile organic compound (VOC) globally, with the total emissions reaching approximately 500 TgC per year, exceeding those of the total anthropogenic VOCs (Guenther et al., 2012; Huang et al., 2017). The high atmospheric reactivity makes it a key precursor for tropospheric ozone ($O_3$) and secondary organic aerosol, both of which significantly impact air quality and climate (Paulot et al., 2012; Lin et al., 2013). In particular, the effect is pronounced in urban environments due to the interactions between isoprene and anthropogenic emissions (Xu et al., 2015).

Terrestrial vegetation is the primary source of atmospheric isoprene, and the emissions are influenced by plant species, geographical locations, and environmental conditions (Guenther et al., 1994; Guenther et al., 1993). Urban landscapes show remarkable diversity in isoprene production, exemplified by stark differences between tree species. While urban greenspace offers numerous benefits, it also emerges as a notable contributor to urban isoprene (Ma et al., 2022). The emissions are highly sensitive to meteorological conditions (Wang et al., 2024a). The combination of climate warming and urbanization lead to intensified urban heat, which in turn boosts isoprene emissions from greenspace (Li et al., 2024; Pfannerstill et al., 2024). In addition, studies have shown that a portion of urban isoprene may stem from motor vehicles, the contribution of which varies by location and season (Borbon et al., 2001). However, vehicle emissions of isoprene do not necessarily increase with growing vehicle population, due to stringent emission controls in many cities. This further complicates the task of accurately simulating the concentrations and trends of urban isoprene. While isoprene may also be emitted from other anthropogenic sources, such as petrochemical activities and coal combustion, the amounts are generally small compared to biogenic emissions, especially in warm seasons (Peron et al., 2024).

Modeling and measurement deficiencies remain a serious concern in isoprene research across multiple disciplines. For example, the Model of Emissions of Gases and Aerosols from Nature (MEGAN) estimates vegetation emissions based on theoretical



relationships with meteorology and vegetation dynamics. This model significantly
underestimates isoprene emissions from urban greenspace when it is driven by coarse
resolution (e.g., >30 m) satellite-derived vegetation data (Ma et al., 2019; Ma et al.,
2022). It is also difficult for current chemical transport models to accurately simulate
isoprene concentrations, mainly resulting from the grid resolution and uncertainties in
isoprene emissions, vertical dispersion rates and oxidation parameterization schemes
(Arneth et al., 2011; Guenther et al., 2012). Local vegetation surveys and emission
factor measurements can be made to improve model performance. However, the work
is challenging and the outcomes often point out additional uncertainties (Seco et al.,
2022; Wang et al., 2024b). While isoprene measurements have demonstrated reliability
in atmospheric chemistry research, the temporal and spatial coverage remains
suboptimal. Given these constraints, there are insufficient robust isoprene data available
over climatic timescales (e.g., several decades) to reveal the drivers of long-term trends.
To confront this dilemma, we developed a generalized physics-informed neural
network based on a residual Multi-Layer Perceptron with a transfer training strategy to
reproduce/predict ambient isoprene concentrations. The model was trained by a
comprehensive set of isoprene data observed at ten sites in China (a total of ~65,000
hourly data) and validated by a total of ~8,500 hourly and daily data at six overseas
sites (Table S1). The model was verified for its ability to predict isoprene with limited
sizes of observational data and understand intricate relationships between isoprene and
influencing factors. The model was then used to predict future trends of isoprene and
the resulting $O_3$ variations in different climate scenarios. This study enhances our
understanding of the responses of ambient isoprene concentrations to emissions and
meteorology, and has the potential to inform urban planning and air quality policies in
the warming climate.



## 2 Data and Models

### 2.1 Isoprene Data and Deep Learning Model

A total of over 72,000 hourly (and a small fraction of daily) data of isoprene concentrations in the daytime (06:00–20:00 local time) of warm seasons (May–October) were compiled from 16 sites worldwide. Around 88% the data was from different parts of China, and the remainder was from North America, Amazonia, India, and the UK (see Table S1). To ensure comparability, we included only online measurements, excluding offline sampling and analysis methods. While inter-instrument bias might exist, the isoprene variability within each site was expected to be much larger than any plausible inter-instrument bias. Moreover, this will not influence the analysis of isoprene variations at individual sites.

The residual multi-layer perceptron architecture (ResMLP) was employed to approximate the complicated responses of isoprene concentrations to input features, which was coupled with a physics-informed neural network, thereby PINN-ResMLP. This approach integrated domain knowledge by enforcing monotonicity constraints between isoprene and its major sources (e.g., vegetation and traffic emissions), thus ensuring physically consistent predictions. These constraints were implemented directly in the model's loss function, which combined terms for data fitting, monotonicity regularization, and network structure penalties. As a fully data-driven model, ResMLP may learn patterns that are inconsistent with physical laws. Incorporating expert knowledge and physical constraints into the model can guide the learning processes. In this study, we stipulated that isoprene concentrations were positively correlated with the biogenic and traffic emission sources. This relationship therefore can be expressed as:

$$\frac{\partial\,ISOP}{\partial\,VI} > 0 \tag{1}$$

$$\frac{\partial\,ISOP}{\partial\,BC_{traffic}} > 0 \tag{2}$$





where ISOP represents isoprene concentrations; VI is vegetation index derived from
Leaf Area Index (LAI) and Normalized Difference Vegetation Index (NDVI) (see Text
S1); and $BC_{traffic}$ is traffic emissions of black carbon. To satisfy this priori knowledge,
we developed PINN-ResMLP to constrain the model. The optimization objective of
PINN-ResMLP included data item loss ( $\mathcal{L}_{data}$ ), physical inconsistency loss
($\mathcal{L}_{monotonicity}$), and additional structural loss ($\mathcal{L}_{structure}$). Meanwhile, the L2 norm of
the network parameters, namely adding the sum of the squares of all network weights
(parameters) to the loss function, could effectively regularize and prevent overfitting in
PINN-ResMLP. Finally, the total function was formulated as:
$$\mathcal{L} = \mathcal{L}_{data} + \alpha \cdot \mathcal{L}_{monotonicity} + \beta \cdot \mathcal{L}_{structure} \tag{3}$$

$$\mathcal{L}_{data} = \frac{1}{2N} \sum_{i=1}^{N} \left[ \left( ISOP_{obs,i} - ISOP_{pred,i} \right)^2 + \left| ISOP_{obs,i} - ISOP_{pred,i} \right| \right] \tag{4}$$

$$\mathcal{L}_{monotonicity} = \frac{1}{N} \sum_{i=1}^{N} \left[ 1 - \frac{sign\left(\frac{\partial ISOP}{\partial VI}\right) + sign\left(\frac{\partial ISOP}{\partial BC_{traffic}}\right)}{2} \right] \tag{5}$$

$$sign(\theta) = \begin{cases} -1, & \theta < 0 \\ 1, & \theta \geq 0 \end{cases} \tag{6}$$

$$\mathcal{L}_{structure} = \sum_{i=1}^{M} (W_i^2 + b_i^2) \tag{7}$$

where the $\alpha$ and $\beta$ are trade-off parameters; N is the number of training samples; $i$
represents a certain sample; $ISOP_{obs}$ and $ISOP_{pred}$ are observed and predicted
isoprene concentrations, respectively; M is the number of layers in PINN-ResMLP.
Predictor variables were selected to capture key sources and sinks of isoprene, including
VI (see Text S1), meteorological parameters (e.g., temperature, solar radiation), and
black carbon emitted from traffic ($BC_{traffic}$) as a proxy for anthropogenic emissions. Full
variable definitions and sources are provided in Table S2.
To address data scarcity at some sites, we implemented a supervised transfer learning
strategy. The PINN-ResMLP was pre-trained on data-rich sites and fine-tuned with
limited data from target sites (see Table S3). Three training strategies were adopted: (1)
transfer learning (T), where models were pre-trained on data from other sites and fine-



tuned on the target site; (2) no-transfer (NT), where models were trained solely on target
site data; and (3) mixed training (MIX) using combined data from all sites. Model
performance was evaluated using four-fold cross-validation (Table S3) and metrics
including normalized mean absolute error (NMAE) and $R^2$. Comparisons were made
against standard machine learning algorithms, such as Random Forest (RF), extreme
gradient boosting (XGB), and support vector machine (SVM). All algorithms were
optimized using extensive grid search (see hyperparameters in Table S4).

**2.2 Attribution of Long-term Isoprene Trends and $O_3$ Projections**

The PINN-ResMLP model was also used to quantify the contributions of different
factors to the long-term trends of isoprene concentrations at three sites in Hong Kong
and London using a scenario-based approach. Using the historical data of
meteorological parameters, VI and $BC_{traffic}$ as the input of the PINN-ResMLP$_T$ model,
we predicted the summertime (June to August) isoprene concentrations in Hong Kong
and London for the period of 1990–2023 (base scenario). In order to reveal the impacts
of the major drivers on the isoprene variations, we also predicted the isoprene
concentrations by fixing the temperature, VI and $BC_{traffic}$ as their averages over the
above period one by one (controlled scenarios). The differences in the predicted
isoprene (isoprene$_{diff}$) between the base and controlled scenarios depicted the isoprene
trends induced by the individual factors. The isoprene$_{diff}$ was then compared between
different time periods, e.g., the first and last 17 years and the first, middle and last
decades. Besides, the coefficient of variation (CV) was calculated for the predicted
isoprene concentrations in all the scenarios over the period of 1990–2023. The CV
differences between the base and controlled scenarios indicate how the changes in
temperature, VI and traffic emissions (represented by $BC_{traffic}$) increased or decreased
the variations in isoprene concentrations.
Additionally, future isoprene concentrations at the Hong Kong site were projected for
2030–2100 based on temperature changes under different climate scenarios developed
by the Coupled Model Intercomparison Project Phase 6 (CMIP6), while keeping other
influencing factors constant. Briefly, we used the temperature data from four Shared



Socioeconomic Pathways (SSPs), including SSP126 (low forcing), SSP245
(intermediate forcing), SSP370 (medium-high forcing), and SSP585 (high forcing), and
held the other influencing factors constant. The averages of outputs from four Coupled
Model Intercomparison Project Phase 6 (CMIP6) models (ACCESS-CM2, CMCC-
ESM2, MPI-ESM1-2-HR, and GFDL-ESM4) were adopted (Xu et al., 2024).
Using the future profiles of temperature and isoprene as constraints, we simulated $O_3$
concentrations as a function of isoprene and temperature using a zero-dimensional box
model under different $NO_x$ reduction scenarios. The Framework for 0-D Atmospheric
Modeling (F0AM) incorporating Master Chemical Mechanism v3.3.1 was used to
simulate $O_3$ under different sets of temperatures and isoprene concentrations (Lyu et al.,
2024). The model was constrained by the average diurnal profiles of air pollutants
(excluding $O_3$) and meteorological parameters observed in the summer of 2023 at the
Hong Kong_TC site, except that the daytime average temperature changed from 22 °C
to 38 °C in 2 °C intervals and the daytime average isoprene concentrations varied in the
range of 0.15–1.8 ppbv in intervals of 0.15 ppbv. The $O_3$ isopleths were depicted using
the simulation results for 108 temperature-isoprene settings. Additionally, the above
simulations were repeated in different scenarios of $NO_x$ reduction, i.e., 49.7% and 82.6%
under the SSP370 and SSP126, respectively. It is worth noting that the diurnal profiles
of other $O_3$ precursors, such as VOCs and carbon monoxide, were kept unchanged
throughout all the simulations.



## 3 Results and Discussion

### 3.1 Simulating Isoprene Concentrations Using PINN-ResMLP$_T$ Model

The isoprene concentrations averaged over the respective observation periods varied significantly from 0.15 ppbv in Wuhan to 2.79 ppbv in New Delhi (Figure 1 and Figure S1), due to the differences in sampling periods, climatic conditions, and vegetation type and density. We noticed that most cities have experienced an increase in greenspace in the last 20 years, and there existed significant differences in greenspace coverage and its recent trends between the cities (e.g., Hong Kong versus London). The high vegetation cover appeared to explain the elevated levels of isoprene in South China and Amazonia. Temperature also had a strong effect on isoprene concentrations, as indicated by the Pearson correlation coefficient (R) between hourly isoprene and temperature at individual sites, i.e., 0.41–0.72. The temperature and temperature variation were spatially non-uniform, implying its inconsistent roles in affecting biogenic isoprene emissions. Additionally, anthropogenic emissions might have made significant contributions to isoprene in New Delhi, given the high nocturnal levels and the peak in evening rush hours (Figure S2). This was also mentioned in a previous study (Tripathi et al., 2022). The Weather Research and Forecasting model with Chemistry, which theoretically takes these factors into account, was used to simulate the isoprene concentrations. However, substantial divergences were noted between the simulated and observed values at hourly or even daily resolution (Figure S3), demonstrating the challenge for chemical transport models in accurately simulating isoprene (Morichetti et al., 2022; Wang et al., 2024b).

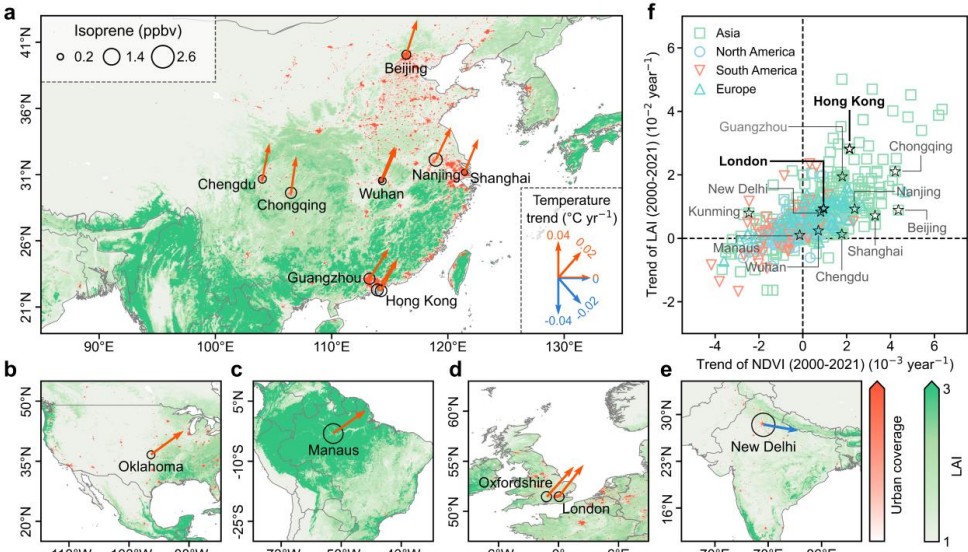

**Figure 1.** Geographical distribution of the isoprene sampling sites. (**a-e**) Locations of isoprene measurement sites in China (**a**), North America (**b**), Amazonia (**c**), United Kingdom (**d**), and India (**e**). The direction of arrows represents the trend of temperature from 1990 to 2023, and the size of the circle is proportional to isoprene concentration. (**f**) Trends of LAI and NDVI from 2001 to 2021 in major cities around the world.

We then examined the isoprene prediction ability of various machine learning algorithms with three training strategies: T, NT, and MIX (see Section 2.3). Overall, the model utilizing the NT training strategy exhibited higher fitting accuracy than the one employing the MIX training strategy (Figure 2). This suggests that training with data from different sites might introduce additional noises, due to the differences in isoprene emission dynamics. Particularly, isoprene emissions are highly sensitive to local vegetation profiles. While the ResMLP model with the NT training strategy (ResMLP$_{NT}$) performed moderately among all the algorithms, the performance was improved by incorporating the T strategy. Specifically, the ResMLP$_T$ outperformed the other algorithms at 8 out of 10 sites, with the decrease in NMAE of 1%–5% and increase in $R^2$ of 0.01–0.07. The results indicated that the ResMLP$_T$ model effectively exploited the implicit prior knowledge from the pre-training data to guide isoprene prediction at the target sites. Importantly, the pre-trained parameters were fine-tuned using limited



sizes of local data, which adapted the model to local isoprene emission dynamics
without requiring region-specific vegetation profiles, such as vegetation types and
corresponding emission factors. It is worth noting that the size of the retraining data at
the validation sites was ~30% of all the data. Thus, the model's good performance at
the validation sites demonstrated its utility in data-scarce regions. Furthermore, with
the incorporation of PINN, the PINN-ResMLP$_T$ showed a better understanding of the
real target-feature relationships with more interpretable prediction results. The model
performance was further improved, as indicated by the highest $R^2$ values across all the
sites (Figure 2).
Next, we also validated the PINN-ResMLP$_T$ model by applying it to predict isoprene
concentrations at the overseas sites (Figure 1). The model was pre-trained with the
complete dataset from all the sites in China, which was further fine-tuned with 70% of
the data at the individual target sites and validated with the remaining data. Compared
to the suboptimal model, the PINN-ResMLP$_T$ model significantly improved the
prediction of isoprene, especially at New Delhi and Manaus, with the increase in $R^2$
(decrease by NMAE) by 0.07 (25%) and 0.08 (13%), respectively (Figure 2). This
demonstrated the model's broad applicability. Moreover, the model outperforms many
other methods in predicting isoprene. For instance, the root mean square error of the
PINN-ResMLP$_T$ at Manaus was 0.17 ppbv, compared to 0.95 ppbv for the early attempt
in Cross-track Infrared Sounder retrieval (Fu et al., 2019). This superior performance
establishes the PINN-ResMLP$_T$ as our best choice. In fact, we would expect more
accurate predictions if the model were pre-trained by a wider range of field
measurement data from various regions.





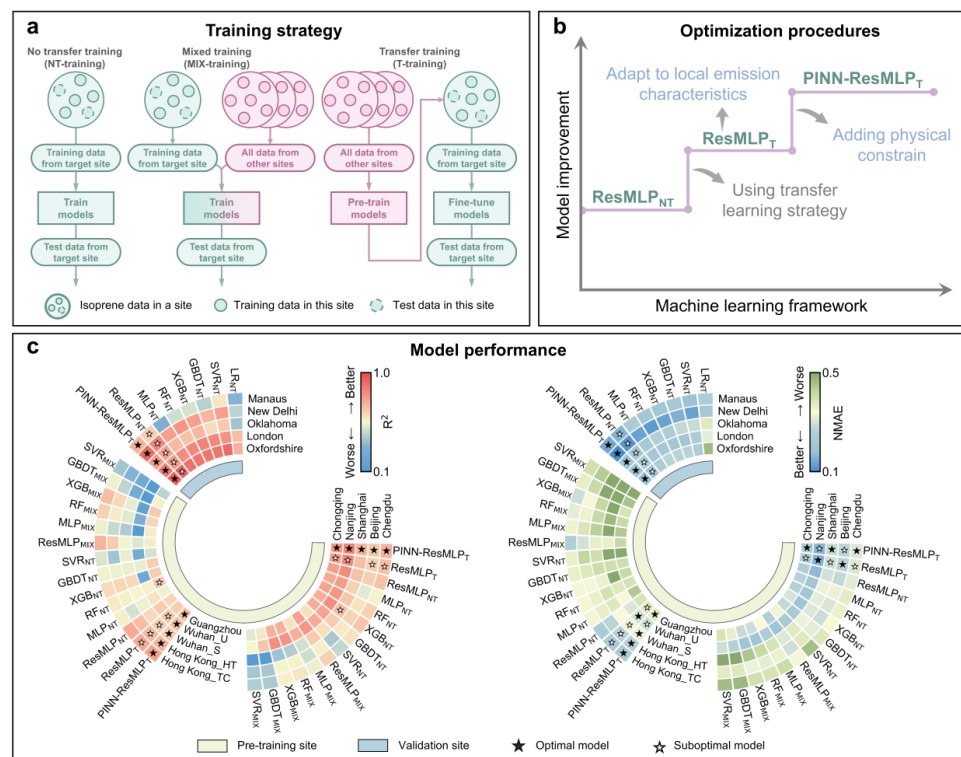

**Figure 2.** Schematic of the machine learning framework for predicting isoprene concentration.
(**a**) A sketch of the training strategy. (**b**) The improvement of the machine learning framework.
(**c**) Comparisons of statistical performance across different algorithms in individual sites. See
methods for the full names of the algorithms.



**3.2 Main Factors Influencing Urban Isoprene Concentrations**


Further, a feature importance method based on the the SHapley Additive exPlanations
(SHAP) values was employed to explore the prediction results (Figure 3). While we
prefer to present it for individual sites, the feature importance of VI and $BC_{traffic}$ was
not calculated for the Chinese sites except a suburban site in Hong Kong (HK_TC), due
to the low temporal resolutions of VI and $BC_{traffic}$ data and the short isoprene
observation periods. Here, we discuss the drivers of short-term (2-4 years) and long-
term (over 10 years) isoprene concentration variations, separately.
With the VI and $BC_{traffic}$ remaining relatively stable in the short term, the model
indicated that temperature, radiation, surface pressure, and soil water vapor were the
most significant drivers of short-term isoprene variations, and their average relative
importance was 18.8%, 11.9%, 11.3%, and 8.1% across the all the Mainland China sites,
respectively. In addition, evaporation from vegetation transpiration and relative
humidity also played significant roles in affecting isoprene concentrations at the Wuhan
suburban site and Beijing urban site, respectively. The model also effectively captured
the target-feature relationships. In China, the predicted isoprene concentrations
increased with temperature below ~35 °C, above which a decline occurred at some sites.
A typical example was the response in Chongqing with frequent occurrence of high
temperature extremes (Figure S4). High temperatures suppress vegetation emissions
due to a reduction in enzyme activity and substrate availability while accelerating the
photochemical oxidation loss of isoprene. A similar pattern was observed in the
response of isoprene concentrations to radiation in China, especially at extremely high
levels. Such nonlinear responses are critical to the parameterization of isoprene
emissions in numerical models. Notably, recent studies have revealed substantial
uncertainties in the MEGAN model's performance under extreme heat conditions
(Wang et al., 2024a). In contrast, our data-driven machine learning approach effectively
captures these complex, nonlinear relationships between isoprene concentrations and
environmental predictors, offering a promising pathway to refine and optimize
parameterization schemes in chemical transport models. In addition to the close



relationships with temperature, solar radiation affects the hydroxyl radical
concentrations and therefore can significantly increase chemical loss of isoprene. In
contrast, these phenomena were not observed in London, because of lower temperatures
and weaker solar radiations. Overall, our transfer learning model reasonably reflected
the isoprene-meteorology relationships.

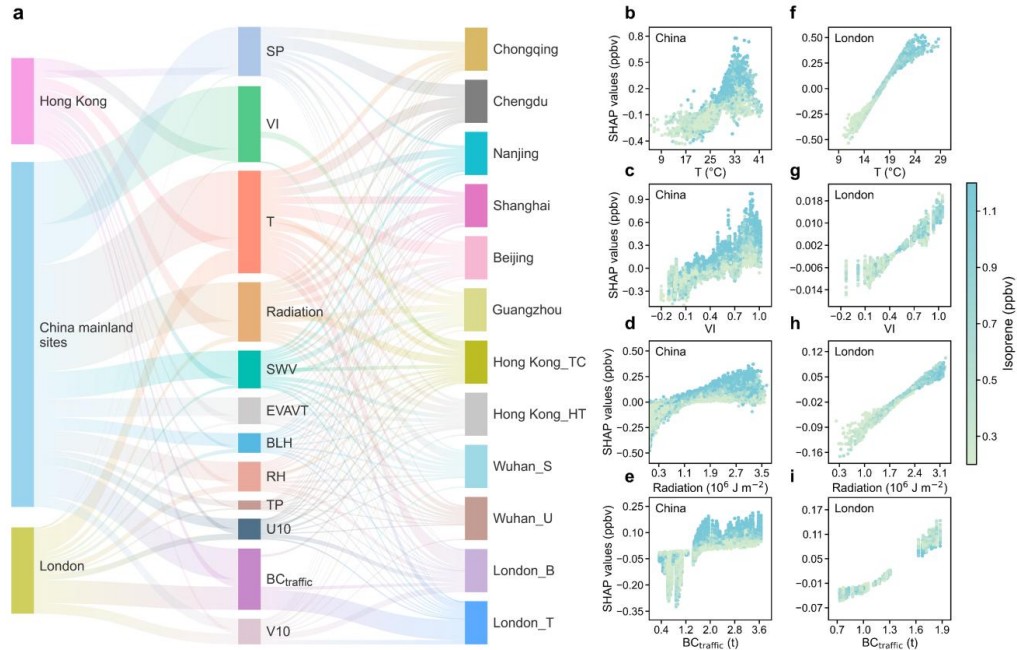


**Figure 3.** Modeling explainable results at each site based on SHAP value. (**a**) Feature
importance for isoprene concentrations at individual sites. (**b-i**) The SHAP dependence plots of
major influencing variables averaged at the Chinese sites (**b-e**) and London sites (**f-i**).
Further, the long-term isoprene observations in London and Hong Kong offer an
opportunity to examine how the evolutions of VI and $BC_{traffic}$ affected isoprene
variations on a climatic scale. As two prominent international cities, London and Hong
Kong are characterized by distinct climatic zones: London experiences a mid-latitude
temperate maritime climate, whereas Hong Kong is influenced by a low-latitude
subtropical monsoon climate. This climatic differentiation is reflected in their
predominant vegetation types, with temperate deciduous trees being prevalent in



London and evergreen broad-leaved trees dominating the landscape in Hong Kong.
Furthermore, the trajectories of urbanization and air pollution management have
evolved differently in each city, shaped by their unique environmental and socio-
economic contexts. Here, we focus on the results at two London sites and a Hong Kong
site, where long term data was available. Radiation, VI and temperature were the most
predominant influencing factors at the suburban site in Hong Kong. In contrast, the
relative importance of VI was low at the two London sites. Over the period of 2000–
2021, Hong Kong has experienced a notable increase in NDVI ($2.1 \times 10^{-3}$ year$^{-1}$) and
LAI ($2.8 \times 10^{-2}$ year$^{-1}$), while the rate was much lower in London, i.e., $0.9 \times 10^{-3}$ year$^{-1}$
for NDVI and $0.9 \times 10^{-2}$ year$^{-1}$ for LAI. Additionally, the significant difference in VI
importance between Hong Kong and London might also be attributed to the different
strength of vegetation emissions across latitudes (Guenther et al., 2006; Guenther et al.,
2012). Instead, BC$_{traffic}$ (temperature) ranked the first at the traffic (suburban) site in
London, followed by other meteorological factors (Figure 3). While the meteorological
impacts were not surprising, isoprene correlated well with the BC$_{traffic}$ emissions and
observed benzene at the London traffic site (Figure S5), thereby the high relative
importance of BC$_{traffic}$. This is consistent with the previous studies on traffic emissions
of isoprene in London (Borbon et al., 2001; Von Schneidemesser et al., 2011). As
constrained using the PINN, the SHAP values for isoprene concentrations of VI and
BC$_{traffic}$ showed a monotonic increasing trend.





**3.3 Factors Driving Long-term Trends of Isoprene**

The model was also used to build the time series data of daytime isoprene concentrations at a daily resolution over a climatically relevant period (1990–2023) at the three sites with long-term but incomplete isoprene data. The comparison between the geographically distinctive London and Hong Kong offers a rare opportunity to examine the different drivers of isoprene trends. As shown in Figure 4, the predicted isoprene concentrations were in good agreement with the observations, with the $R^2$ of 0.68–0.83 and NMAE of 21%–27%. It's worth noting that the observational data was missing for 50%–67% of the dates at the three sites. This underscored the model's effectiveness in retrieving historical isoprene concentrations from limited observation data.

Over the past 34 years, the isoprene concentrations at the Hong Kong site have shown an increasing trend with the rate of 18.1 pptv year$^{-1}$, as have the temperature and VI. In contrast, traffic emissions have been significantly reduced since 1998, due to the effective human interventions. The trend of the predicted isoprene correlated strongly with the SHAP values of VI (R = 0.95) and moderately with that of temperature (R = 0.63). By fixing the variables one by one, we determined the variations in factor contributions to isoprene concentrations, which were then compared between different time periods. It was found that urban greenspace emerged as the dominant factor impacting Hong Kong's isoprene levels, causing a rise in isoprene concentrations of 290 pptv between the last 17 years and the first 17 years. Meanwhile, the contribution of climate warming was 51 pptv, while the traffic contribution was minor. Moreover, without changes in urban greenspace, the coefficient of variation (CV) of annual average isoprene concentrations would decrease by 70.5%, in comparison to the decrease of 12.0% and 6.0% in absence of changes in climate warming and traffic emissions, respectively. This reiterated the significant impacts of urban greenspace on the variations and trends of isoprene concentrations.

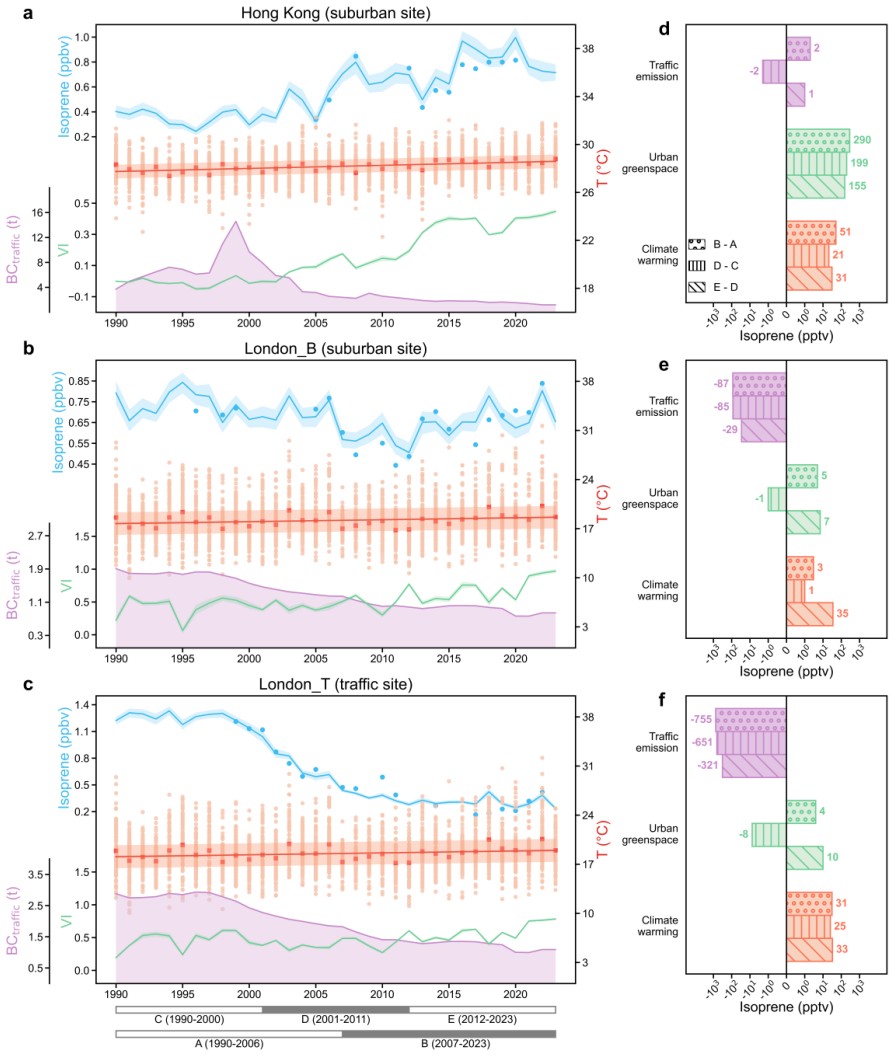

**Figure 4.** Long-term trends of the summertime isoprene and the drivers. (**a-c**) Variations of isoprene concentrations (blue lines for simulated, blue dots for observed), temperature (T), urban greenspace (VI) and traffic emissions (BC$_{traffic}$) in Hong Kong (**a**) and London (**b** and **c**). The red dots and red line represent temperature and the fitted trend for the mean temperature, respectively. The bands represent 95% confidence intervals. (**d-f**) Changes in isoprene concentrations caused by climate warming, urban greenspace and traffic emissions in Hong Kong (**d**) and London (**e** and **f**) during different periods.





In contrast, the isoprene concentrations in London were lower in the last 17 years. Climate warming would have increased the isoprene concentrations by 31 pptv compared to those in the first 17 years at the traffic site, while the impact at the suburban site and the effects of urban greenspace at both sites were negligible. Interestingly, traffic emissions accounted for 87 pptv and 755 pptv of isoprene reduction at the suburban site and the traffic site, respectively. This was likely attributed to stringent traffic emission controls, as indicated by the significant downward trend of $BC_{traffic}$. The effect was more pronounced during the first two thirds of the study period (1990–2011). Specifically, the traffic-related isoprene reduction was 85 pptv from the first (1990–2000) to the second decade (2001–2011) at the suburban site, in comparison to the 29 pptv between the second and the last decade (2012–2023). Actually, the observed isoprene concentration correlated moderately (R = 0.60) with $BC_{traffic}$ from 1990 to 2011 based on their annual averages. This suggests that traffic emission controls also affected isoprene concentrations even in non-urban areas. Despite the higher VI in London, the increasing rate (1.2 year$^{-1}$ at the suburban site and 0.7 year$^{-1}$ at the traffic site) was lower than that in Hong Kong (1.6 year$^{-1}$). Additionally, the weak effects of urban greenspace might be also due to the relatively low emission strengths of high-latitude vegetation (Guenther et al., 2006). Moreover, the impact of climate warming became evident in the last decade (2012–2023) at the suburban site in London and, together with urban greenspace, reversed the isoprene reduction that would otherwise have been achieved by traffic emission controls. This aligned with the accelerated temperature rise from 2011 onwards (Figure S6), which was also reported elsewhere (Cao et al., 2021). From the perspective of variations in annual isoprene concentrations, the CV at the traffic site primarily resulted from changes in traffic emissions. At the suburban site, it would decrease by 32.4% and 14.0% if temperature and traffic emissions did not change.

Overall, our results demonstrate a tale of two cities: similarities and differences in drivers of long-term isoprene trends. Temperature-driven increases in isoprene concentrations were revealed in both cities, especially in the last decade, underscoring



the universal impacts of climate warming on vegetation emissions. However, the disparities in green space changes and probably different biogenic isoprene emission strengths between the two cities led to the different effects of VI on isoprene variations. Additionally, the isoprene variations over the 34 years have been more influenced by traffic emissions in London, although both cities have implemented stringent vehicle emission controls. While the reasons remain to be explored, we did not identify any correlation between the observed isoprene and $BC_{traffic}$ (or benzene) in Hong Kong, even at a traffic site (Figure S5).



### 3.4 Future Projections for Isoprene and $O_3$

A significant issue associated with increasing isoprene levels in a warming climate is the potential for elevated ground-level $O_3$ pollution. We used the temperature from the latest CMIP6 multi-model ensemble forecasts to predict isoprene concentrations from 2030 to 2100 in Hong Kong under four IPCC's shared socio-economic pathway (SSP) scenarios, while the other factors were kept constant. As shown in Figure 5a, the temperature is expected to increase by 0.71–3.60 ℃ from 2030 to 2100. The model indicated that the daytime average concentration of isoprene would increase by 87–530 pptv (15%–87%) by 2100 (Figure 5b). The changes are on the same magnitude as the previous estimates that isoprene emissions will increase by 21%–57% by the end of this century relative to the 1990–2010 levels (Cao et al., 2021; Sanderson et al., 2003).

Further, we simulated the $O_3$-isoprene-temperature relationships in Hong Kong (as an example) using future temperatures and isoprene concentrations while fixing the other air pollutants and meteorological conditions at present levels. The simulated $O_3$ increased markedly with the rise in temperature and isoprene concentrations (Figure 5c-5f). The $O_3$ concentration would increase by up to 1.7 folds by 2100 under the SSP585 scenario of temperatures and isoprene. An increase in the combined risk of heat and $O_3$ exposure could be expected. To explore the approach of alleviating the adverse impact of $O_3$-isoprene-temperature synergy, we proposed additional scenarios by cutting anthropogenic $NO_x$ emissions. With the $NO_x$ reduction from the current to different SSPs levels, the $O_3$ concentrations would increase and decrease under low and high isoprene-temperature conditions, respectively (Figure 5d-5f). This inconsistent variation is due to the evolution of $O_3$ formation regime with the rising temperatures and isoprene. It is worth noting that more ambitious $NO_x$ reduction would result in greater $O_3$ benefits. For example, $O_3$ would decrease in a much wider range of temperature and isoprene when $NO_x$ is reduced under SSP126. The $O_3$ growth by 2100 would be only 1.2 folds in the SSP585 scenario of temperatures and isoprene. Therefore, substantial reduction in anthropogenic $NO_x$ would effectively address the synergy between temperature, isoprene and $O_3$.



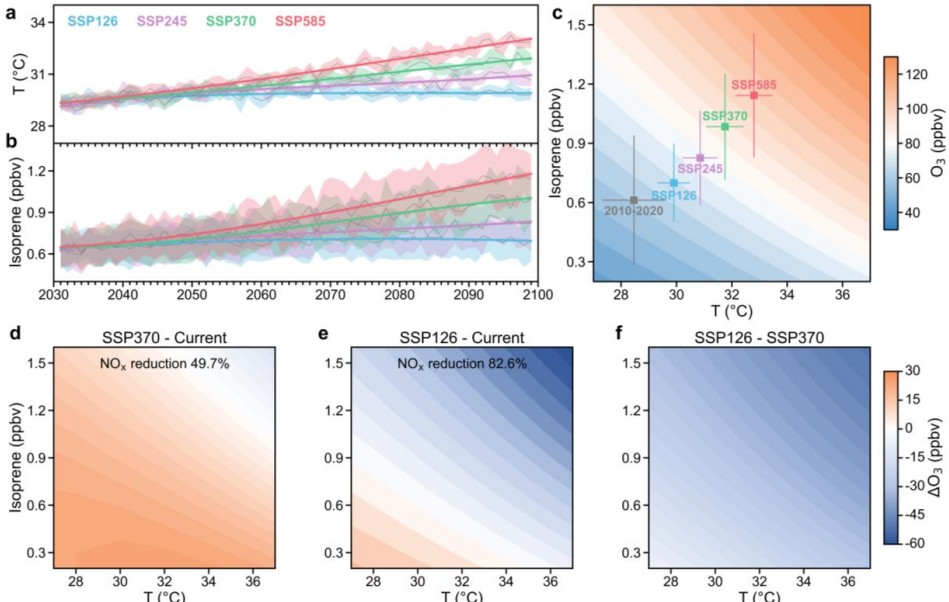

**Figure 5.** Projected temperature, isoprene and $O_3$-isoprene-temperature relationships under different climate scenarios. (**a-b**) Projections of the summertime daytime air temperature (**a**) and isoprene concentrations (**b**) during 2030-2100 in Hong Kong. The shaded areas represent the 25th to 75th percentile of the estimated isoprene concentration and temperature for each SSP. (**c-f**) Responses of simulated $O_3$ concentrations to temperature and isoprene under abundant-$NO_x$ (**c**) and reduced-$NO_x$ (**d-f**) conditions. The squares represent the projected $O_3$ concentrations at specific temperatures and isoprene levels, with error bars indicating the standard deviation of isoprene concentrations and temperatures.



**4 Conclusions**
As one of the most reactive and abundant VOC, isoprene plays a significant role in
shaping urban air quality. We developed an explainable deep transfer learning
framework to predict isoprene concentrations and elucidate the underlying drivers of
their variability. Our model outperformed conventional approaches, effectively
capturing the spatial heterogeneity of isoprene concentrations through localized fine-
tuning. Leveraging this framework, we quantified the relative importance of factors
influencing isoprene concentrations across numerous sites in China and internationally.
The contrasting cases of Hong Kong and London highlight how isoprene dynamics
were shaped by distinct local drivers, underscoring the need to tailor air quality
management strategies to specific urban contexts. Despite the anticipated increase in
biogenic emissions in a warming climate, our findings caution against reducing urban
greenspace solely based on isoprene-related concerns. Instead, mitigating global
warming emerges as a crucial strategy for managing isoprene's air quality impacts, as
evidenced by the strong isoprene–temperature relationships observed. For $O_3$
abatement, coordinated control of $NO_x$ emissions appears effective in reducing the
contribution of isoprene to $O_3$ formation. Moreover, the differing responses of isoprene
to VI between Hong Kong and London suggest that informed tree species selection can
serve as an alternative urban planning measure. Traffic emissions may also remain a
significant source of urban isoprene in cities lacking stringent vehicle emission controls
and should be addressed accordingly. Overall, this study provides novel insights into
isoprene emissions and chemistry, air quality impacts, and practical mitigation
strategies. Nonetheless, limitations persist, particularly regarding the comprehensive
representation of emissions and chemical loss processes, which are discussed in Text
S2.



**Acknowledgments**

This work was supported by National Key Research and Development Program (2023YFC3709304), Hong Kong Research Grants Council via the General Research Fund (HKBU 15219621, HKBU 15209223), National Natural Science Foundation of China (42575120, 42293322), Public Policy Research Funding Scheme (2023.A2.059.23C), the Youth Fund Project of the Sichuan Provincial Natural Science Foundation (24NSFSC2988), and the Fundamental Research Funds for the Central Universities (YJ202313).

**Open Research**

Meteorology data for 1990-2023 at each site were obtained from the hourly ERA5 reanalysis dataset (Hersbach et al., 2023); The CMIP6 model outputs can be accessed at https://pcmdi.llnl.gov/CMIP6/; The NDVI data from 1990-2022 are available at https://doi.org/10.3334/ORNLDAAC/2187; The GLASS LAI V5 and V6 products are downloaded from https://www.glass.hku.hk/download.html and https://www.geodata.cn/main/, respectively.

**Author contributions**

X.L. and N.W. conceived the study. S.L. developed the methodology. Data collection was performed by S.L., N.W., X.L., Z.S., X.H., T.L., H.W., M.L., J.G., N.C., G.S., Y.Z., C.P., Z.L., C.T., and X.L. Data analysis was conducted by S.L., X.L., and N.W. N.W. and X.L. led the investigation and supervision. Visualization was completed by S.L. and N.W. The original draft was written by S.L., N.W., and X.L. All authors, including F.Y., Z.S., X.H., and A.G., contributed to reviewing and editing the manuscript.

**Competing Interests**

The authors declare that they have no known competing financial interests or personal relationships that could have appeared to influence the work.





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
