# Peer review of "Deciphering Isoprene Variability Across Dozen of Chinese and Overseas Cities Using Deep Transfer Learning"

_EGUsphere, 2025_

## Referee Comment (RC1)

**Comments:**

This study employs machine learning techniques to investigate the patterns and driving factors underlying the fluctuations in isoprene levels — a crucial precursor to surface ozone — using extensive historical datasets. The analysis demonstrates good agreement with previous short-term modeling results, while also emphasizing that vegetation expansion and temperature increases linked to climate change are key drivers of long-term variability. Moreover, by extending the model projections to the year 2100 and integrating them with a detailed chemical box model, the researchers estimated future surface ozone levels. Their results indicate a significant rise in ozone concentrations if NOx emissions are not effectively controlled. Importantly, these conclusions were obtained using a data-driven approach that differs from conventional atmospheric chemical transport models, highlighting the robustness and novelty of the findings. I recommend acceptance for publication in ACP after minor revisions.

- 1. The superior performance of T-training suggests that reliable simulations can be achieved for grid cells or cities with existing isoprene observations. Furthermore, could this method also be applied to regions lacking monitoring data?
- 2. The decrease in vehicle emissions seems to significantly influence isoprene concentrations at traffic sites in London, as further supported by the comparison with benzene. Have prior studies presented similar evidence or discussed this effect?
- 3. Please include the calculation formulas for R2 and NMAE.
- 4. Some input variables have varying spatial resolutions. How were these differences addressed, and what were the primary criteria for selecting these variables?
- 5. In the PINN-ResMLP model, an additional loss term was incorporated into the training process. Was the training stable across different sites, and how did the model

loss change accordingly?

6. Please ensure that the use of abbreviations is consistent between the main text and

the appendix tables, such as Radiation vs. SSRD and U10 vs. u10.

7. Please supply additional details regarding the isoprene measurement instruments

used at each site.

8. Please include a definition of SHAP in the Methods section.

9. In the simulation of future scenarios with NOx reduction, please specify the basis

or reference for the assumed reductions of 49.7% and 89.2%.

10. Please add the full names of LAI and NDVI in the caption of Figure 1.

11. Additionally, since Figure 2 includes multiple algorithm abbreviations, it would be

beneficial to define them in the caption for improved clarity.

12. Line 162: provide the full term for R2.

13. Line 129: add a reference.

---

## Author Comment (AC1)

**Reviewer #1**

**General comments:**

This study employs machine learning techniques to investigate the patterns and driving factors underlying the fluctuations in isoprene levels — a crucial precursor to surface ozone — using extensive historical datasets. The analysis demonstrates good agreement with previous short-term modeling results, while also emphasizing that vegetation expansion and temperature increases linked to climate change are key drivers of long-term variability. Moreover, by extending the model projections to the year 2100 and integrating them with a detailed chemical box model, the researchers estimated future surface ozone levels. Their results indicate a significant rise in ozone concentrations if $NO_x$ emissions are not effectively controlled. Importantly, these conclusions were obtained using a data-driven approach that differs from conventional atmospheric chemical transport models, highlighting the robustness and novelty of the findings. I recommend acceptance for publication in ACP after minor revisions.

**Response:** We sincerely thank the reviewer for the positive and encouraging evaluation of our work. We appreciate the recognition of the study's novelty, the robustness of the data-driven framework, and the relevance of our findings regarding long-term isoprene variability and its impactions for ozone ($O_3$). We have carefully addressed all issues raised by the reviewer and revised the manuscript accordingly. We believe that these changes have further improved the clarity and quality of the paper. Below, we provide the point-by-point responses to each comment, with our replies highlighted in blue and the corresponding revisions marked in red.

**Specific comments:**

*1. The superior performance of T-training suggests that reliable simulations can be achieved for grid cells or cities with existing isoprene observations. Furthermore, could this method also be applied to regions lacking monitoring data?*

**Response:** We thank the reviewer for this insightful question. T-training performs best in regions with existing isoprene observations, as these provide the necessary

information to train the model effectively (Gupta et al., 2021; Theodoris et al., 2023). For regions lacking monitoring data, the model could still be applied using a transfer-learning or domain-adaptation approach, leveraging patterns learned from observationally constrained regions. However, we note that predictions in such regions may carry higher uncertainties due to the absence of local measurements, and this should be taken into account when interpreting the results (Wells et al., 2020).

**Reference:**

Gupta, V., Choudhary, K., Tavazza, F., et al.: Cross-property deep transfer learning framework for enhanced predictive analytics on small materials data, Nat. Commun., 12, 6595, 10.1038/s41467-021-26921-5, 2021.

Theodoris, C. V., Xiao, L., Chopra, A., et al.: Transfer learning enables predictions in network biology, Nature, 618, 616-624, 10.1038/s41586-023-06139-9, 2023.

Wells, K. C., Millet, D. B., Payne, V. H., et al.: Satellite isoprene retrievals constrain emissions and atmospheric oxidation, Nature, 585, 225-233, 10.1038/s41586-020-2664-3, 2020.

*2. The decrease in vehicle emissions seems to significantly influence isoprene concentrations at traffic sites in London, as further supported by the comparison with benzene. Have prior studies presented similar evidence or discussed this effect?*

**Response:** Our analysis indicates that traffic emissions have had a stronger influence on long-term isoprene variations at the London traffic site. To further support this finding, we examined the correlation between benzene and $BC_{traffic}$. As shown in the Figure R1, $BC_{traffic}$ exhibits a strong correlation with both benzene (R = 0.85) and isoprene (R = 0.79), which further confirms the role of traffic emissions in shaping long-term isoprene variations in London_T site.

Evidence from previous studies is consistent with our findings. Khan et al. (2018) demonstrated that anthropogenic emissions from traffic in London can contribute substantially to ambient isoprene, in some cases even exceeding biogenic contributions during daytime. Similar observations have been reported across Western Europe cities (including London and Paris), where traffic-related isoprene has been identified as a

non-negligible component of urban VOC budgets (Borbon et al., 2023). These earlier results corroborate our interpretation of the traffic influence on isoprene trends in London.

[Figure]

Figure R1. Correlation analysis of monthly isoprene concentrations with benzene and BC_traffic at London traffic site.

**Reference:**

Borbon, A., Dominutti, P., Panopoulou, A., et al.: Ubiquity of Anthropogenic Terpenoids in Cities Worldwide: Emission Ratios, Emission Quantification and Implications for Urban Atmospheric Chemistry, J. Geophys. Res.-Atmos., 128, 10.1029/2022jd037566, 2023.

Khan, M. A. H., Schlich, B. L., Jenkin, M. E., et al.: A Two-Decade Anthropogenic and Biogenic Isoprene Emissions Study in a London Urban Background and a London Urban Traffic Site, Atmosphere, 9, 10.3390/atmos9100387, 2018.

*3. Please include the calculation formulas for $R^2$ and NMAE.*

**Response:** We have added the calculation formulas for the coefficient of determination ($R^2$) and the normalized mean absolute error (NMAE) in the Methods section to improve clarity and reproducibility.

**Please see the revisions in Line 168-171 of the manuscript:** "NMAE and $R^2$ are calculated as follows:

$$NMAE = \frac{\frac{1}{N}\sum_{i=1}^{N}\left|ISOP_{obs,i} - ISOP_{pred,i}\right|}{\overline{ISOP}_{obs}} \tag{8}$$

$$R^2 = 1 - \frac{\sum_{i=1}^{N}\left(ISOP_{obs,i} - ISOP_{pred,i}\right)^2}{\sum_{i=1}^{N}\left(ISOP_{obs,i} - \overline{ISOP}_{obs}\right)^2} \tag{9}$$

where $\overline{ISOP}_{obs}$ represents the mean value of $ISOP_{obs}$."

*4. Some input variables have varying spatial resolutions. How were these differences addressed, and what were the primary criteria for selecting these variables?*

**Response:** Input variables were selected based on the MEGAN framework to represent the primary sources, sinks, and atmospheric processes affecting isoprene. Biogenic emissions were captured through vegetation-related factors summarized as a vegetation index (VI) derived from LAI and NDVI. Meteorological variables that influence isoprene emission activity and chemical reactivity, such as temperature, solar radiation, humidity, wind, and boundary layer height, were included. Traffic emissions, identified as the most important anthropogenic source, were also selected as an input variable. To ensure consistency among all variables, datasets with different spatial resolutions (e.g., 0.1° and 0.25°) were resampled to a uniform resolution of 0.1° using bilinear interpolation.

**Please see the revisions in Line 151-156 of the manuscript:** "Predictor variables were selected to capture key sources and sinks of isoprene, including VI (see Text S1), meteorological parameters (e.g., temperature, solar radiation), and black carbon emitted from traffic ($BC_{traffic}$) as a proxy for anthropogenic emissions. To ensure consistency among all variables, datasets with different spatial resolutions (e.g., 0.1° and 0.25°) were resampled to achieve a uniform resolution of 0.1° using bilinear interpolation. Full variable definitions and sources are provided in Table S2."

*5. In the PINN-ResMLP model, an additional loss term was incorporated into the training process. Was the training stable across different sites, and how did the model*

*loss change accordingly?*

**Response:** The PINN-ResMLP model training was stable across all sites. During training, the total loss and each individual loss component (including the additional term) consistently decreased and converged within the expected number of epochs (Figure R2). Minor fluctuations were observed early in training due to differences in site-specific data distributions, but these quickly stabilized (Figure R2). Overall, the incorporation of the additional loss term did not compromise training stability and effectively guided the model to capture the prior knowledge.

[Figure]

Figure R2. Training curves of the PINN-ResMLP model showing the total loss and individual loss components, including reconstruction loss, monotonicity loss, and structure loss.

*6. Please ensure that the use of abbreviations is consistent between the main text and the appendix tables, such as Radiation vs. SSRD and U10 vs. u10.*

**Response:** Thank you for pointing this out. We have carefully checked all abbreviations in the main text, figures, and appendix tables, and have revised them to ensure consistency throughout the manuscript.

*7. Please supply additional details regarding the isoprene measurement instruments used at each site.*

**Response:** Additional details about the isoprene measurement instruments used at each site have been added to the Supporting Information (Table S5).

**Please see the revisions of the Supporting Information:**

Table S5. Summary of site characteristics and instrumentation for isoprene measurements.

| Site | Latitude | Longitude | Instrument | Reference |
|------|----------|-----------|------------|-----------|
| Beijing | 40.05° | 116.42° | GC–FID/MS | Zhang et al., (2024) |
| Chengdu | 30.66° | 104.04° | Synspec GC955-611/811 | Song et al., (2018) |
| Chongqing | 29.62° | 106.51° | Synspec GC955-611/811 | Wang et al., (2024) |
| Guangzhou | 23.08° | 113.37° | AC-GCMS1000 | Tong et al., (2024) |
| Hong Kong_TC | 22.29° | 113.94° | GC-PID | Wang et al., 2017) |
| Hong Kong_HT | 22.22° | 114.26° | GC-PID | Wang et al., (2017) |
| Nanjing | 32.12° | 118.96° | GC-MS/FID | Li et al., (2024) |
| Shanghai | 31.17° | 121.43° | GC-FID | Wang et al., (2020) |
| Wuhan_U | 30.53° | 114.37° | GC-FID/MS | Wang et al., (2014) |
| Wuhan_S | 30.60° | 114.28° | GC-FID/MS | Wang et al., (2014) |
| London_T | 51.45° | 0.07° | Perkin Elmer Ozone Precursor Analysers | Derwent et al., (2014) |

| | | | | |
|---|---|---|---|---|
| London_B | 51.52° | 0.16° | Perkin Elmer Ozone Precursor Analysers | Derwent et al., (2014) |
| Oklahoma | 36.60° | -97.49° | PTR-MS | Liu et al., (2021) |
| Manaus | -3.10° | -59.99° | PTR-MS | Nascimento et al., (2021) |
| Oxfordshire | 51.46° | -1.20° | GC-PID | Ferracci et al., (2020) |
| New Delhi | 28.45° | 77.28° | PTR-TOF-MS 8000 | Tripathi et al., (2022) |

*8. Please include a definition of SHAP in the Methods section.*

**Response:** We have added a detailed description of SHAP (Shapley Additive Explanations) in the Methods section, including its calculation principle.

**Please see the revisions in lines 172-179 of the manuscript: "**Finally, the Shapley Additive Explanations (SHAP) approach (Lundberg et al., 2020) was used to quantify the contributions of input variables to model predictions. SHAP values allow us to assess the impact of each factor on isoprene concentrations. The calculation is defined as follows:

$$\varphi_i = \sum_{S \subseteq K\{i\}} \frac{|S|! \, (K - |S| - 1)!}{|K|!} [f(S \cup \{i\}) - f(S)] \qquad (10)$$

where $K$ represents the set of all features, $S \subseteq K\backslash\{i\}$ denotes a feature subset that excludes feature $i$, $|S|$ is the size of subset $S$, $f(S)$ is the model's prediction under feature subset $S$, and $f(S \cup \{i\}) - f(S)$ is the marginal contribution of feature $i$.

**Reference:**

Lundberg, S. M., Erion, G., Chen, H., et al.: From local explanations to global understanding with explainable AI for trees, Nat. Mach. Intell., 2, 56-67, 10.1038/s42256-019-0138-9, 2020.

*9. In the simulation of future scenarios with NO$_x$ reduction, please specify the basis or*

*reference for the assumed reductions of 49.7% and 89.2%.*

**Response:** In this study, the assumed $NO_x$ reductions of 49.7% and 89.2% are derived from the emission trajectories in the Shared Socioeconomic Pathways (SSPs), as quantified by the Global Change Assessment Model (GCAM). GCAM provides future anthropogenic emission projections under different SSP narratives, reflecting varying strengths of air pollution control policies—strong controls in SSP1 and SSP5, medium controls in SSP2, and weak controls in SSP3 and SSP4 (Lou et al., 2023). The reduction ratios used in our simulations were calculated directly from GCAM-based SSP emission datasets, which include sector-resolved $NO_x$ emissions from energy supply and demand, industry, transportation, land use, waste, and solvent use (Rogelj et al., 2018). These GCAM-derived $NO_x$ trajectories form the basis for the reduction scenarios adopted in the box-model simulations.

Reference:

Lou, S., Shrivastava, M., Ding, A., et al.: Shift in Peaks of PAH-Associated Health Risks From East Asia to South Asia and Africa in the Future, Earth Future, 11, e2022EF003185, https://doi.org/10.1029/2022EF003185, 2023.

Rogelj, J., Popp, A., Calvin, K. V., et al.: Scenarios towards limiting global mean temperature increase below 1.5 °C, Nat. Clim. Change, 8, 325-332, 10.1038/s41558-018-0091-3, 2018.

*10. Please add the full names of LAI and NDVI in the caption of Figure 1.*

**Response:** We have added the full names of LAI (Leaf Area Index) and NDVI (Normalized Difference Vegetation Index) to the caption of Figure 1.

**Please see our revisions in lines 255-257 of the manuscript: "**Trends of LAI (Leaf Area Index) and NDVI (Normalized Difference Vegetation Index) from 2001 to 2021 in major cities around the world.**"**

*11. Additionally, since Figure 2 includes multiple algorithm abbreviations, it would be beneficial to define them in the caption for improved clarity.*

**Response:** Thank you for the suggestion. All algorithm abbreviations appearing in

Figure 2 have now been defined in the figure caption to enhance clarity for readers.

**Please see our revisions in lines 297-300 of the manuscript: "**RF, XGB, GBDT, SVM, and LR represent Random Forest, eXtreme Gradient Boosting, Gradient Boosting Decision Tree, Support Vector Machine, and Linear Regression, respectively.**"**

*12. Line 162: provide the full term for $R^2$.*

**Response:** We have replaced "$R^2$" with its full term, "coefficient of determination ($R^2$)," at the first occurrence in the manuscript.

**Please see our revisions in lines 162-165 of the manuscript: "**Model performance was evaluated using four-fold cross-validation (Table S3) and metrics including normalized mean absolute error (NMAE) and coefficient of determination ($R^2$).**"**

*13. Line 129: add a reference*

**Response:** A relevant reference has been added at Line 129 to support the statement.

**Please see the revisions in lines 128-129 of the manuscript: "**Incorporating expert knowledge and physical constraints into the model can guide the learning processes (Zhu et al., 2024).**"**

**Reference:**

Zhu, B., Ren, S., Weng, Q., et al.: A physics-informed neural network that considers monotonic relationships for predicting NOx emissions from coal-fired boilers, Fuel, 364, 131026, https://doi.org/10.1016/j.fuel.2024.131026, 2024.

---

## Author Comment (AC2)

**Reviewer #2**

**General comments:**

The manuscript presents an explainable deep transfer learning framework (PINN-ResMLP) to predict urban isoprene concentrations and attribute their variability across Chinese and international cities. It further explores long-term drivers in Hong Kong and London (1990–2023) and projects future isoprene and ozone responses under CMIP6/SSP scenarios, including $NO_x$-control sensitivity. The study fills an important gap: robust isoprene prediction without detailed local emission inventories or explicit chemistry, and interpretable attribution that links meteorology, greenspace, and traffic to observed and modeled trends. The approach is timely and impactful for urban air quality management in a warming climate. The integration of physics-informed constraints with transfer learning is a notable strength, as is the explicit discuss ability (SHAP-based) of model predictions. The Hong Kong–London contrast is compelling and policy-relevant.

**Response:** We sincerely thank Reviewer #2 for the thorough and positive evaluation of our work. We greatly appreciate the recognition of the novelty and impact of our study, particularly the explainable deep transfer learning framework (PINN-ResMLP), its ability to predict isoprene concentrations without detailed local emission inventories, and the interpretability provided by SHAP analysis. We also thank the reviewer for emphasizing the relevance of the Hong Kong–London comparison and the importance of integrating physics-informed constraints with transfer learning. The constructive feedback and supportive comments are highly encouraging and have helped us further clarify and refine the manuscript. Below, we provide our point-by-point responses to each comment, with our replies highlighted in blue and the corresponding revisions marked in red.

**Specific comments:**

*1. Equations (5–6) use sign functions over partial derivatives. Please clarify how the gradients with respect to inputs are computed for monotonicity (e.g., via automatic*

*differentiation), and whether local monotonicity is enforced pointwise or globally. Also specify α and β values and sensitivity.*

**Response:** We thank the reviewer for this comment. In our PINN-ResMLP model, the gradients with respect to input features (VI and $BC_{traffic}$) are computed using PyTorch's automatic differentiation (autograd). Monotonicity is enforced pointwise, ensuring that the derivative of the model output with respect to each input at each data point satisfies the expected monotonicity constraint. During training, we experimented with different values for the loss weight coefficients α and β in the multi-loss function. Specifically, α was tested in [0.01, 0.1, 1] and β in [0.0001, 0.001, 0.01, 0.1]. We found that α = 1 and β = 0.0001 yielded the best performance in terms of capturing monotonicity without degrading predictive accuracy. Additionally, we corrected the implementation of the sign function in Equations (5) and (6). The corrected form is:

$$\mathcal{L}_{monotonicity} = \frac{1}{N}\sum_{i=1}^{N}\left[-\frac{sign\left(\frac{\partial ISOP}{\partial VI}\right) + sign\left(\frac{\partial ISOP}{\partial BC_{traffic}}\right)}{2}\right] \qquad (5)$$

$$sign(\theta) = \begin{cases} -1, & \theta < 0 \\ 0, & \theta \geq 0 \end{cases} \qquad (6)$$

*2. $\mathcal{L}_{structure}$ is defined as sum of squared weights per layer ($W_i^2 + b_i^2$). Are there any architectural constraints (e.g., skip connections in ResMLP, layer widths) chosen to improve stability? Include a small ablation (ResMLP vs. ResMLP+PINN vs. PINN alone) if possible.*

**Response:** To improve training stability, our ResMLP architecture incorporates residual (skip) connections and carefully chosen layer widths. These design choices help mitigate vanishing/exploding gradient issues and ensure robust convergence. The model is trained using the Adam optimizer with a weight decay term to further regularize the network and stabilize training. While we did not perform a formal ablation study, our results show that incorporating the PINN framework improves predictive accuracy and enforces physical constraints (Figure 2), highlighting the effectiveness of combining ResMLP with physics-informed constraints.

*3. For overseas sites, you fine-tune on 70% and validate on 30%. Clarify whether the split preserves temporal ordering (to reduce leakage) and whether performance is robust to different splits (report variance across splits).*

**Response:** For the overseas validation experiments, due to the limited data size (~1,000 samples), the data split did not preserve temporal ordering. Instead, we performed cross-validation to assess model robustness. The results consistently show that the PINN-ResMLP$_T$ framework improves predictive accuracy across different sites, demonstrating the stable performance of the model.

*4. The authors showed that WRF-Chem performed poorly in isoprene simulations. Provide configuration details (chemistry mechanism, emissions, resolution, boundary conditions) and whether the MEGAN parameterization and land-use were tuned to urban greenspace. This contextualizes the performance gap and its causes (e.g., grid dilution, canopy-scale processes).*

**Response:** In this study, we used the Weather Research and Forecasting model with Chemistry (WRF-Chem, version 3.7) to simulate urban isoprene concentrations. Meteorological initial and lateral boundary conditions were provided by the NCEP FNL dataset at $1° × 1°$ resolution, and Four-Dimensional Data Assimilation (FDDA) was applied to improve the meteorological fields. The Noah land surface model and the MM5 Monin–Obukhov surface layer scheme were used to represent land–atmosphere exchange processes, while the planetary boundary layer was simulated using the YSU scheme. Gas-phase chemistry and aerosol processes were represented using the CBMZ mechanism and the MOSAIC module, respectively. Biogenic VOC emissions were calculated using the Model of Emissions of Gases and Aerosols from Nature (MEGAN v2.1; Guenther et al., 2012). The vegetation-related static inputs were updated using MODIS PFT (MCD12Q1) and LAI (MCD15A2H) products. Anthropogenic emissions were obtained from the updated 2020-based MEIC inventory for regions within China and the MIX inventory (Li et al., 2017) for regions outside China, both at $0.25° × 0.25°$ resolution and including major emission sectors (transportation, industry, power plants, residential, and agriculture). These configurations of WRF-Chem have been

successfully applied in our previous studies (Huang et al., 2020; Huang et al., 2021; Wang et al., 2021), indicating a reliable performance across China.

We note that the MEGAN and land-use datasets used in this study could not capture BVOC emissions from urban greenspace (which requires high-resolution data, e.g., 10 m × 10 m) or canopy-scale processes, as the MODIS satellite product is limited to 500 m × 500 m. Combined with the relatively coarse model resolution (WRF-Chem grid resolution: 25 km × 25 km), these limitations may lead to grid dilution of urban vegetation signals and underestimation of peak isoprene emissions, helping to explain the performance gap observed between WRF-Chem and our PINN-ResMLP framework.

**Reference:**

Huang, X., Ding, A., Wang, Z., et al.: Amplified transboundary transport of haze by aerosol – boundary layer interaction in China, Nature Geoscience, 13, 428-434, 10.1038/s41561-020-0583-4, 2020.

Huang, X., Ding, A. J., Gao, J., et al.: Enhanced secondary pollution offset reduction of primary emissions during COVID-19 lockdown in China, National Science Review, 8, 10.1093/nsr/nwaa137, 2021.

Wang, N., Xu, J., Pei, C., et al.: Air Quality During COVID-19 Lockdown in the Yangtze River Delta and the Pearl River Delta: Two Different Responsive Mechanisms to Emission Reductions in China, Environ. Sci. Technol., 55, 5721-5730, 10.1021/acs.est.0c08383, 2021.

*5. State whether SHAP is computed on the fine-tuned model per site, the background dataset used, and whether interaction SHAP was explored (temperature × radiation) to reflect coupled sensitivities.*

**Response:** Thank you for the valuable suggestion. In our study, SHAP values for the Chinese sites were computed based on the pre-trained model, while for Hong Kong and London, SHAP values were computed on the fine-tuned neural network model for each site, using the corresponding site's training dataset as the background dataset. Regarding interaction SHAP values (e.g., temperature × radiation), we did not calculate them in the current manuscript. The widely used DeepSHAP implementation in the

official "shap" package (v0.46.0, based on DeepExplainer) only computes marginal (main-effect) SHAP values and does not provide exact pairwise or higher-order interaction terms. Exact SHAP interaction values, as implemented in TreeSHAP for tree-based models, are currently not available for deep neural networks in any mature, computationally tractable form. The fundamental reason is that tree models have discrete decision paths that allow precise attribution of output changes to arbitrary feature coalitions, whereas neural networks exhibit highly nonlinear, continuous interactions across all features simultaneously (Janzing et al., 2020; Zern et al., 2023). Although some existing approximation approaches (e.g., Integrated Hessians) can estimate interactions between paired features in neural networks (Janizek et al., 2021), in this study we focused on the main SHAP effects to identify the key drivers of isoprene variability. We agree that exploring feature interactions is important and plan to investigate this in future work using alternative methods.

**Reference:**

Janizek, J. D., Sturmfels, P., and Lee, S.-I.: Explaining explanations: axiomatic feature interactions for deep networks, 22, Article 104, 2021.

Janzing, D., Minorics, L., and Bloebaum, P.: Feature relevance quantification in explainable AI: A causal problem, Proceedings of the Twenty Third International Conference on Artificial Intelligence and Statistics, Proceedings of Machine Learning Research 2020.

Zern, A., Broelemann, K., and Kasneci, G.: Interventional SHAP Values and Interaction Values for Piecewise Linear Regression Trees, Proceedings of the AAAI Conference on Artificial Intelligence, 37, 11164-11173, 10.1609/aaai.v37i9.26322, 2023.

*6. For future projections, please explicitly acknowledge that future greenspace, urban form, and anthropogenic emissions will also evolve.*

**Response:** We thank the reviewer for this comment. In this work, we adopt a pseudo-global-warming (PGW)–based approach to isolate the effect of climate warming on isoprene emissions and the consequent $O_3$ responses. PGW directly impose selected changes (e.g. temperature changes) in the climate system onto a historical regional

climate simulation by modifying the initial and boundary conditions (Brogli et al., 2023). Following this framework, our future simulations were designed to vary only temperature while holding other precursors and environmental drivers fixed, thereby quantifying the chemical sensitivity of isoprene and $O_3$ to warming alone. Accordingly, the future projections do not account for potential changes in greenspace, urban form, or other anthropogenic emissions. We acknowledge that these factors may evolve over time and that incorporating them could further refine future predictions. We have added a note in the revised manuscript to explicitly clarify this limitation.

**Please see our revisions in lines 219-227 of the manuscript:** "It is worth noting that the diurnal profiles of other $O_3$ precursors, such as VOCs and carbon monoxide, were kept unchanged throughout all the simulations. Meanwhile, our future projections are designed to isolate the chemical response of $O_3$ to changes in temperature and isoprene and do not explicitly incorporate potential future changes in greenspace, urban morphology, or other anthropogenic emissions. Although these factors are expected to evolve under urban development and climate mitigation pathways, the present analysis focuses on quantifying the impacts of climate warming on isoprene emissions and the consequent $O_3$ responses."

**Reference:**

Brogli, R., Heim, C., Mensch, J., et al.: The pseudo-global-warming (PGW) approach: methodology, software package PGW4ERA5 v1.1, validation, and sensitivity analyses, Geosci. Model Dev., 16, 907-926, 10.5194/gmd-16-907-2023, 2023.

*7. There are two "the" in line 280.*

**Response:** The redundant "the" in line 280 has been removed in the revised manuscript.